# Scaling of Rotational Constants

**DOI:** 10.3390/molecules29245874

**Published:** 2024-12-12

**Authors:** Denis S. Tikhonov, Colin J. Sueyoshi, Wenhao Sun, Fan Xie, Maria Khon, Eva Gougoula, Jiayi Li, Freya Berggötz, Himanshi Singh, Christina M. Tonauer, Melanie Schnell

**Affiliations:** 1Deutsches Elektronen-Synchrotron DESY, Notkestr. 85, 22607 Hamburg, Germany; 2Department of Chemistry, Amherst College, Amherst, MA 01002-5000, USA; 3Institute of Physical Chemistry, Christian-Albrechts-Universität zu Kiel, 24118 Kiel, Germany; 4Department of Physics, Universität Hamburg, 22607 Hamburg, Germany

**Keywords:** rotational constants, scaling factors, density functional theory

## Abstract

This manuscript introduces the concept of scaling factors for rotational constants. These factors are designed to bring computed equilibrium rotational constants closer to experimentally fitted ground-state-averaged rotational constants. The parameterization of the scaling factors was performed for several levels of theory, namely DF-D*n*/def2-*m*VP (DF=B3LYP,PBE0, n=3(BJ),4, m=S,TZ), PBEh-3c, and r${\,}^{2}$SCAN-3c. The obtained scaling factors systematically improved the consistency between the theoretical and experimental rotational constants.

## 1. Introduction

Rotational spectroscopy, consisting of microwave (MW) [1,2], millimeter wave (MMW) [3], and terahertz (THz) [4,5,6] spectroscopies, is a powerful high-resolution experimental technique that provides unprecedented structural sensitivity for the different structural (including conformational) and constitutional isomerisms of molecules [7,8,9], as well as for isotopic substitutions and various large-amplitude vibrational motions [10], such as proton transfer [11], internal rotation [12], inversion [13], and even movements of an entire molecule across another molecule’s surface [9]. This kind of spectroscopic technique can be used in various applications, from monitoring the chemical composition of mixtures [14] and reactions [3,15] to detecting atmospheric [16,17] and interstellar molecular species [8,18].

In the zeroth approximation, the rotational spectrum of a molecule is given by the rigid rotor model, which in the case of non-linear molecules is parametrized using three rotational constants, which are denoted as *A*, *B*, and *C* or, equivalently, as $B_{a}$, $B_{b}$, and $B_{c}$, respectively [19,20,21]. These constants, usually expressed in MHz, are related to the moments of inertia $I_{\alpha }$ along the given α-th principal axis of the molecule through the following expression [19,20]:(1)$$B_{\alpha }=\frac{\hbar }{4\pi I_{\alpha }}\;,$$
where α=a,b,c is the given principal axis, $\hbar =1.05\times 10^{-34}$ J·s is the reduced Planck constant, and the moment of inertia is given as
(2)$$I_{\alpha }=\sum_{n=1}^{N}m_{n}\cdot r_{\alpha n}^{2}\;,$$
where *n* enumerates the atoms in the molecule, *N* is the overall number of atoms in the molecule, $m_{n}$ is the mass of the *n*-th atom, and $r_{\alpha n}$ is the distance of the *n*-th atom to the α-th principal axis. In the experiment, the so-called vibrationally averaged rotational constants are obtained, usually in the ground vibrational state, and they are commonly denoted as $A_{0}$, $B_{0}$, and $C_{0}$ [21].

The initial structural assignment of a spectroscopically observed species is usually conducted based on quantum–chemical (QC) calculations. The candidate structures are optimized at a chosen level of theory, usually with a dispersion-corrected density functional theory (DFT) calculation, and then the theoretical rotational constants are compared with the experimentally determined ones [22]. However, such a comparison does not always yield an unambiguous assignment of the molecular structures, and the reason for this is two-fold. First of all, the optimized structure corresponds to the so-called equilibrium geometry with the corresponding equilibrium rotational constants $A_{e}$, $B_{e}$, and $C_{e}$, in which all vibrational effects are absent [21,23]. Therefore, the experimental $A_{0}$, $B_{0}$, and $C_{0}$ values and theoretical $A_{e}$, $B_{e}$, and $C_{e}$ are not equal due to the vibrational anharmonic shifts, which usually expand the molecular size, thus increasing the moments of inertia (Equation (Equation 2)) and consequently decreasing the corresponding rotational constant (Equation (Equation 1)) [24]. In other words, it is generally expected that $B_{0,\alpha }\leq B_{e,\alpha }$. The second reason is the quality of the QC approximation, which can distort the equilibrium structure due to the complicated underestimation and/or overestimation of various intra- and intermolecular chemical bonds and non-covalent interactions. This systematic error does not have a preferred shift of the rotational constant values with respect to their experimental counterparts and thus can be of any type [21].

In this work, we propose to systematically improve the inconsistency between experimental ground-state-averaged rotational constants ($A_{0}$, $B_{0}$, $C_{0}$) and their theoretical equilibrium counterparts ($A_{e}$, $B_{e}$, $C_{e}$) by applying tabulated scaling factors. Such an approach, where the band shifts due to anharmonic effects and QC approximation failure, has been demonstrated to be fruitful in the case of vibrational (e.g., infrared) spectroscopy [25,26,27,28,29,30,31,32]. Therefore, it is interesting to investigate whether a similar systematic improvement for the lower-frequency spectral range can be achieved as well. First, we will introduce the procedure of the scaling, the fitting model, and the training dataset; then, we will provide the scaling factors, and in the end, we will give an application example using a few recently studied systems with the usage of the PBE0-D3(BJ)/def2-TZVP level of theory.

## 2. Scaling Procedure

We propose to perform the scaling of the theoretical equilibrium constants obtained from QC calculations ($A_{e}^{\mathrm{QC}}$, $B_{e}^{\mathrm{QC}}$, $C_{e}^{\mathrm{QC}}$) with a single global scaling factor, *s*, for a given QC approximation, such that the adjusted constants $A_{s}$, $B_{s}$, and $C_{s}$ are defined as
(3)$$\left\{\begin{array}{c} A_{s}=s\cdot A_{e}^{\mathrm{QC}}\;,\\ B_{s}=s\cdot B_{e}^{\mathrm{QC}}\;,\\ C_{s}=s\cdot C_{e}^{\mathrm{QC}}\;. \end{array}\right.$$
Since the rotational constants are inversely proportional to the moments of inertia (see Equations (Equation 1) and (Equation 2)), such a scaling procedure is effectively equivalent to the global scaling of the atomic coordinates:(4)$$r_{s,n}=\frac{r_{e,n}}{\sqrt{s}}\;,$$
where $r_{e,n}$ and $r_{s,n}$ are the equilibrium and scaled positions of the *n*-th atom in the molecule (see Equation (Equation 2)).

The scale factors are supposed to work as follows. The user optimizes the molecular geometry at a given level of theory to produce the equilibrium rotational constants ($A_{e}^{\mathrm{QC}}$, $B_{e}^{\mathrm{QC}}$, $C_{e}^{\mathrm{QC}}$). Then, a single tabulated scale factor, *s*, is taken. These rotational constants are multiplied with this factor, as given in Equation (Equation 3), to produce scaled rotational constants ($A_{s}$, $B_{s}$, $C_{s}$), which by design should better resemble the experimentally measured values. Then, the comparison of the experimental and scaled rotational constants can be made, e.g., to identify particular conformers in the gas phase (see refs. [7,9,15,22,33]).

To determine and tabulate the scaling factors *s* for given QC approximations, we need a benchmark dataset. In this case, we chose the set of molecules used for obtaining the scaling factors for harmonic frequencies from ref. [34], which were computed using ORCA 5 software [35,36]. From the set of 441 neutral singlet molecules, 174 non-linear molecules with up to 17 atoms with experimentally available $A_{0}$, $B_{0}$, and $C_{0}$ values were selected. Only the relatively high-quality QC approximations were considered here, namely DF-D*n*/def2-*m*VP (DF=B3LYP,PBE0; n=3(BJ),4; m=S,TZ) [37,38,39,40,41,42,43], PBEh-3c [44], and r${\,}^{2}$SCAN-3c [45] levels of theory, while the calculations with the 6-31G basis set were left out from the discussion here. This was performed since rotational spectroscopy, as a high-resolution technique, generally requires more high-quality data for comparison than rotationally unresolved infrared spectroscopy. In addition, MW spectroscopy, as the lowest frequency gas-phase spectroscopic technique to this date, is practically limited by the size of the systems that can be brought into the gas phase in sufficient amounts and by the spectral resolution, as for large systems, the spectra will become dense and uninterpretable, despite the experimental resolution being of the order of $\Delta \nu /\nu ∼10^{-6}--10^{-7}$ in the standard arrangement [46].

As the values of the rotational constants A≥B≥C can differ by orders of magnitude, the metrics that use the absolute deviations between the experimental and theoretical rotational constants are essentially useless, as they will mostly fit the *A*-rotational constants for small-size molecules. Therefore, an advantageous approach is switching to relative values fitting, which was introduced in ref. [34]. In its application to rotational constants, the least-squares problem can be written in the following way:(5)$$\begin{array}{c} {\mathrm{rRSMD}}^{2}\left(s\right)=\\ =\sum_{k=1}^{N_{\mathrm{mol}}}\left({\left(s\cdot \frac{A_{e,k}^{\mathrm{QC}}}{A_{0,k}^{\exp}}-1\right)}^{2}+{\left(s\cdot \frac{B_{e,k}^{\mathrm{QC}}}{B_{0,k}^{\exp}}-1\right)}^{2}+{\left(s\cdot \frac{C_{e,k}^{\mathrm{QC}}}{C_{0,k}^{\exp}}-1\right)}^{2}\right)\to \min\;, \end{array}$$
where rRMSD denotes the relative root mean square deviation (rRMSD) of the rotational constants, *k* enumerates the molecules in the dataset, and $N_{\mathrm{mol}}$ is the total number of molecules in the dataset. The optimal scaling factors are thus given via the following equation [34]:(6)$$s_{\mathrm{opt}}=\underset{s}{\mathrm{argmin}}\left({\mathrm{rRSMD}}^{2}\left(s\right)\right)=\frac{\sum_{k=1}^{N_{\mathrm{mol}}}\left(\frac{A_{e,k}^{\mathrm{QC}}}{A_{0,k}^{\exp}}+\frac{B_{e,k}^{\mathrm{QC}}}{B_{0,k}^{\exp}}+\frac{C_{e,k}^{\mathrm{QC}}}{C_{0,k}^{\exp}}\right)}{\sum_{k=1}^{N_{\mathrm{mol}}}\left({\left(\frac{A_{e,k}^{\mathrm{QC}}}{A_{0,k}^{\exp}}\right)}^{2}+{\left(\frac{B_{e,k}^{\mathrm{QC}}}{B_{0,k}^{\exp}}\right)}^{2}+{\left(\frac{C_{e,k}^{\mathrm{QC}}}{C_{0,k}^{\exp}}\right)}^{2}\right)}\;,$$
where argmin denotes the minimal value of *s*, which minimizes the corresponding function value. The fitting uncertainty of this value is given by the following equation:(7)$$\sigma_{\mathrm{opt}}=\frac{\mathrm{rRSMD}(s_{\mathrm{opt}})}{\sqrt{\sum_{k=1}^{N_{\mathrm{mol}}}\left({\left(\frac{A_{e,k}^{\mathrm{QC}}}{A_{0,k}^{\exp}}\right)}^{2}+{\left(\frac{B_{e,k}^{\mathrm{QC}}}{B_{0,k}^{\exp}}\right)}^{2}+{\left(\frac{C_{e,k}^{\mathrm{QC}}}{C_{0,k}^{\exp}}\right)}^{2}\right)}}\;,$$
where $\mathrm{rRSMD}(s_{\mathrm{opt}})$ is the value of the rRMSD (Equation (Equation 5)), with the optimal scaling factor given in Equation (Equation 6). Note that we do not use weighting with the standard deviations of the experimental fits here because the theoretical calculations have much larger systematic uncertainties that we cannot account for.

## 3. Resulting Scaling Factors

The resulting scaling factors for various levels of theory, as well as the rRMSD values (Equation (Equation 5)) for the unscaled and optimally scaled theoretical rotational constants, are given in Table 1. Several trends can be observed from these results. First, the scaling improved the match between the theory and experiment in most cases, except for B3LYP-D*n*/def2-TZVP (n=3(BJ), 4), which we will discuss later. We also see that the increase in the basis set quality from def2-SVP to def2-TZVP improved the agreement between the experiment and theory in both scaled and unscaled cases. Applying either the D3(BJ) or D4 dispersion correction led to the same scaling factors, which probably points to the equal performance of these corrections. A similar trend was observed for the harmonic frequency scaling factors in ref. [34]. However, the most unexpected yet predictable result is that the optimal scaling factors for the B3LYP-D*n*/def2-TZVP levels of theory were equal to one within the margins of error. This means that the scaling did not significantly improve the predicted rotational constant at this approximation. At the same time, the rRMSD values for the B3LYP-D*n*/def2-TZVP levels are amongst the best in the dataset. Such behavior matches the popularity of these levels of theory for quantum–chemical computations among the rotational spectroscopy community.

## 4. Illustrative Cases

The simplest way to illustrate the robustness and generality of the scaling procedure is to apply this procedure to cases outside the training dataset. For this, we demonstrate the applicability of the obtained scaling factors for a popular quantum–chemical approximation, namely PBE0-D3(BJ)/def2-TZVP [40,41,42]. The geometries of the molecules discussed here were optimized at this level of theory using the ORCA 5 [35,36] software, and then their rotational constants were taken from the calculations. In the case of isotopic substitutions, the rotational constants of the isotopologues were re-computed from the optimized geometries using the UNEX 1.6 software [47]. To demonstrate the numerical performance of the scaling factor, we compared the rRMSD (Equation (Equation 5)) and mean absolute deviations (MADs) of the rotational constants from the experimentally determined values for the given molecules in the case of scaled and unscaled theoretical equilibrium rotational constants. The MAD values were calculated according to the following expression:(8)$$\mathrm{MAD}\left(s\right)=\sum_{k=1}^{N_{\mathrm{mol}}}\left(\left|s\cdot A_{e,k}^{\mathrm{QC}}-A_{0,k}^{\exp}\right|+\left|s\cdot B_{e,k}^{\mathrm{QC}}-B_{0,k}^{\exp}\right|+\left|s\cdot C_{e,k}^{\mathrm{QC}}-C_{0,k}^{\exp}\right|\right)\;.$$

The first illustrative set of molecules included 15 linear top molecules from di- to pentatomic molecules. Since linear molecules have only one rotational constant, they were excluded from the training set, and the scaling effect on these systems will be the most clearly visible. The calculation of the rRMSD and MAD values (Equations (Equation 5) and (Equation 8)) for the linear molecules, thus, included only the *B* rotational constant. The second illustrative set was the case of isotopologues, which, for the simplicity of the analysis, were not included in the training dataset. We chose single-substituted isotopologues of imidazole ($C_{3}H_{4}N_{2}$), which had rotational constants of three singly substituted ${\,}^{13}$C and two singly substituted ${\,}^{15}$N isotopologues available from the literature [48]. The last example was a set of non-covalently bound molecular systems, namely, water–hydrochloric acid clusters $\mathrm{HCl}{\left(H_{2}O\right)}_{n}\;\left(n=2--5,7\right)$, which are examples of hydrogen bond network structures. The rotational constants for these species were taken from refs. [33,49].

We can first take a look at a few exemplary cases of molecular systems from our test dataset, including two linear molecules (HCN and HCCCN), one imidazole ${\,}^{15}$N-substituted isotopologue, namely imidazole-${\,}^{15}$N(1) (nomenclature adapted from ref. [48]), and also the largest of our hydrochloric acid clusters, $\mathrm{HCl}{\left(H_{2}O\right)}_{7}$. The structures of these molecules and their rotational constants are given in Figure 1 and Table 2. As one can see, B3LYP-D3(BJ)/def2-TZVP provided a reasonable estimation of the rotational constants, closer to the experimental values than the unscaled constants at the PBE0-D3(BJ)/def2-TZVP level of theory. However, PBE0-D3(BJ)/def2-TZVP after scaling became as accurate or even more accurate than the B3LYP-D3(BJ)/def2-TZVP-based results. This can be seen by comparing the deviations within the datasets. By looking at the rRMSD and MAD values for the scaled and unscaled rotational constants of these systems at the PBE0-D3(BJ)/def2-TZVP level of theory (Table 3), we observe that the scaling indeed improved the agreement of the theoretical and experimental values. Note that the rRMSD and MAD deviations for the non-covalently bound cluster are larger than for the covalently bound linear molecules and imidazole isotopologues. This might be either due to the less stiff nature of the intermolecular bonds, which allows for larger systematic deviations of the obtained numerical structures from the actual potential energy surface minimum, or due to a less accurate description of the intermolecular interactions in general, which leads to larger systematic errors, or both.

## 5. Conclusions

In this work, we introduced the concept of scaling factors for rotational constants. Applying a single tabulated scaling factor for all rotational constants was effectively equivalent to scaling the molecular size to account for systematic errors in the equilibrium structure due to the quantum–chemical approximation and for absent anharmonic effects. Sets of scaling factors for ten different DFT approximations, namely DF-D*n*/def2-*m*VP (DF=B3LYP,PBE0; n=3(BJ),4; m=S,TZ) and PBEh-3c and r${\,}^{2}$SCAN-3c, were produced from the database of 174 non-linear molecules. The applicability of these scaling factors was illustrated for the PBE0-D3(BJ)/def2-TZVP level of theory in the case of linear molecules, isotopologues, and non-covalently bonded systems. Thus, the application of such scaling factors can be recommended for the more accurate identification of species in rotational spectra and to support the assignment of specific molecular species in complicated broadband rotational spectra.

## Figures and Tables

**Figure 1 molecules-29-05874-f001:**
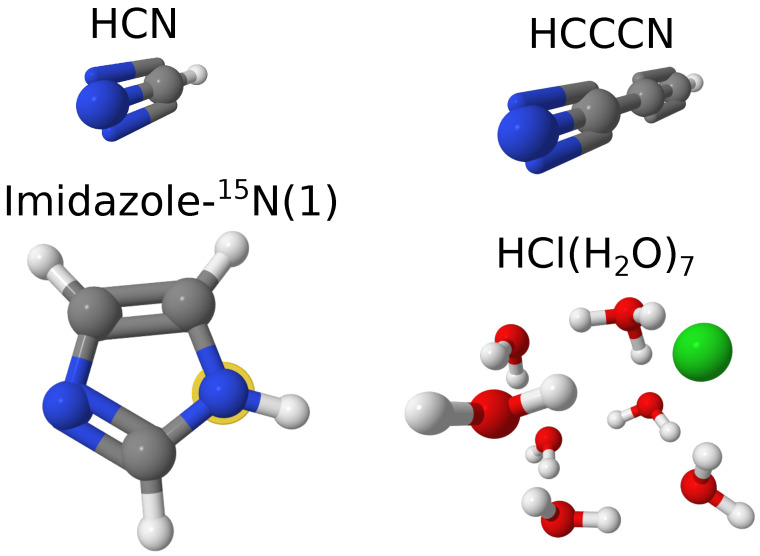
Examplary molecular systems to demonstrate the effect of scaling factors (see main text for details). The atomic color scheme is as follows: hydrogen—white; carbon—gray; nitrogen—blue; oxygen—red; and chlorine—green. The position of isotopic substitution in imidazole is shown with a yellow halo around the atom.

**Table 1 molecules-29-05874-t001:** Optimal scaling factors for rotational constants ($s_{\mathrm{opt}}$) and their uncertainties ($\sigma_{\mathrm{opt}}$) were determined for the listed quantum–chemical approximations with Equations (Equation 6) and (Equation 7), respectively. rRMSD denotes the values of the relative root mean square deviation (Equation (Equation 5)) computed for the training dataset without scaling (s=1) and with an optimized scaling factor ($s=s_{\mathrm{opt}}$).

Method			$s_{\mathrm{opt}}\pm \sigma_{\mathrm{opt}}$	rRMSD(*s*)×100%	
**DF**	**D** n	**Basis**		s=1	$s=s_{\mathrm{opt}}$
B3LYP	D3(BJ)	def2-SVP	1.008±0.001	2.748	2.645
		def2-TZVP	1.000±0.001	2.225	2.225
	D4	def2-SVP	1.008±0.001	2.765	2.656
		def2-TZVP	1.000±0.001	2.244	2.244
PBE0	D3(BJ)	def2-SVP	0.995±0.001	2.633	2.579
		def2-TZVP	0.988±0.001	2.530	2.202
	D4	def2-SVP	0.995±0.001	2.637	2.583
		def2-TZVP	0.988±0.001	2.535	2.206
PBEh-3c			0.987±0.001	3.092	2.778
r${\,}^{2}$SCAN-3c			1.008±0.001	2.689	2.577

**Table 2 molecules-29-05874-t002:** A comparison of experimental and theoretical rotational constants for a few examples of molecular systems: HCN and HCCCN, an imidazole-${\,}^{15}$N(1) isotopologue, and a HCl(H${\,}_{2}$O)${\,}_{7}$ cluster (see Figure 1). “B3LYP” denotes results at the B3LYP-D3(BJ)/def2-TZVP level of theory, and PBE0 denotes results at the PBE0-D3(BJ)/def2-TZVP level of theory. A comparison for all other molecular systems can be found in an Excel spreadsheet in the Appendix A.

Molecular System	A/B/C	Rotational Constant Value [MHz]			
		**Experimental**	**Theoretical**		
			**B3LYP**	**PBE0**	
			s=1	s=1	s=0.988
HCN [50]	*B*	44,316	44,941	44,969	44,415
HCCCN [51]	*B*	4549	4591	4593	4537
Imidazole-${\,}^{15}$N(1) [48]	*A*	9695	9756	9850	9729
	*B*	9188	9218	9271	9157
	*C*	4716	4740	4776	4717
HCl(H${\,}_{2}$O)${\,}_{7}$ [33]	*A*	914	935	947	935
	*B*	737	739	750	740
	*C*	689	709	720	711

**Table 3 molecules-29-05874-t003:** Relative root mean square deviation (rRMSD, Equation (Equation 5)) and mean absolute deviation (MAD, Equation (Equation 8)) values for rotational constants from three illustrative test sets of molecular systems using the PBE0-D3(BJ)/def2-TZVP level of theory. The optimal scaling factor for this method ($s_{\mathrm{opt}}=0.988$) is taken from Table 1.

Dataset	rRMSD(*s*)×100%		MAD(*s*) [MHz]	
	s=1	s=0.988	s=1	s=0.988
Linear molecules	1.0	0.3	33	16
Isotopologues	1.4	0.4	110	26
HCl(H${\,}_{2}$O)${\,}_{n}$	7.9	6.8	132	115

## Data Availability

The Excel sheet containing the data and computations for obtaining the scaling factors is provided in the Appendix A.

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
