# Peer review of "Scaling of Rotational Constants"

_molecules, 2024, doi:10.3390/molecules29245874_

Round 1

Reviewer 1 Report

Comments and Suggestions for Authors

The authors introduce the idea of using scaling parameters to DFT calculations for rotational constants, which can be observed experimentally through spectroscopy. While there is nothing unsound about this work, it is not clear to me that it is also of significant novelty and interest. I think this work would be fine to publish if the authors provide additional motivation and make it clearer what is going on “under the hood” with geometries and how this would work it practice.

In particular, the authors note that the application of a scaling parameter is equivalent to a global scaling of the optimized geometry. While it is true some functionals are known to either systematically over- or underestimate bond lengths, it would be helpful to illustrate or elaborate on how using a separate scaling factor captures anharmonicity about the equilibrium geometry. That point would be much more novel, since most of the time one ends up picking a functional that gives a good geometry anyway (in which case you would be using scaling factors of nearly one, as was seen in the B3LYP/def2-TZVP results). How close were the optimized structures underlying the data in the tables? Would one have picked PBE0 as the “better” choice based on the geometry?

Does the fact that the scaled PBE0 results beat the B3LYP ones mean that while PBE0 has some global error, it has better relative geometry so it can actually be improved?

Do the authors recommend always using these same tabulated values for a functional/basis-set combination, or would one simply scale as necessary for comparing to a given experiment?

How predictive and generalizable is this across other basis sets and functionals? Are there any reasons it might not work for some functionals (such as the B3LYP data) or even make it worse?

Overall, I think this is potentially interesting work, but it requires some additional exploration beyond simply adding a scaling factor to improve agreement to understanding the basic  “when?”, “why?”, and “how?” questions.

Author Response

Thank you for the positive evaluation of our work.

I think this work would be fine to publish if the authors provide additional motivation and make it clearer what is going on “under the hood” with geometries and how this would work it practice.

To provide more details, the following text was added in Section 2:

The scale factors are supposed to work as follows. The user optimizes the molecular geometry at a given level of theory to produce the equilibrium rotational constants (AQCe , BQCe , CQCe). Then, a single tabulated scale factor s is taken. These rotational constants are multiplied with this factor as given in Equation 3 to produce scaled rotational constants (As, Bs, Cs), which by design should better resemble the experimentally measured values. Then, the comparison of the experimental and scaled rotational constants can be made, e.g., to identify particular conformers in the gas phase (see Refs. [7,9,15,22,33]).

While it is true some functionals are known to either systematically over- or underestimate bond lengths, it would be helpful to illustrate or elaborate on how using a separate scaling factor captures anharmonicity about the equilibrium geometry. 

Similar to what is done in IR spectroscopy and with scale factors for the vibrational bands, here we address the anharmonic shifts and systematic errors together with a single scale factor. In principle, separate treatment of the equilibrium rotational constants artifacts and of anharmonic shifts could be done, but it would require high-quality equilibrium geometries and anharmonic shifts to use as the reference data, which is hard to obtain for a dataset of more than 100 molecules. 

That point would be much more novel, since most of the time one ends up picking a functional that gives a good geometry anyway (in which case you would be using scaling factors of nearly one, as was seen in the B3LYP/def2-TZVP results). <...>  Would one have picked PBE0 as the “better” choice based on the geometry?

Does the fact that the scaled PBE0 results beat the B3LYP ones mean that while PBE0 has some global error, it has better relative geometry so it can actually be improved?

Dispersion-corrected B3LYP/def-TZVP calculations, according to our results, are closer not to the experimental equilibrium geometry but to experimental effective r0-geometry, which is convenient but disturbing from the physical point of view. In this case, the seemingly worse performance of dispersion-corrected PBE0/def2-TZVP calculations, in comparison with the experiment, is actually more meaningful and physically sound. From our point of view, the scale factors of 0.988 for PBE0-D3(BJ)/def2-TZVP and PBE0-D4/def2-TZVP indicates that these results are closer to the experimental equilibrium ones, and the residual is the mean value of the anharmonic correction, which seems to be on average 2% of the value of the rotational constant. Therefore, yes, we would argue that it would be better to use, e.g., scaled PBE0-D4/def2-TZVP geometries rather than unscaled B3LYP-D4/def2-TZVP, especially since the agreement with the experiment in the first case, according to tables 1 and 3, is better. 

How close were the optimized structures underlying the data in the tables?

We did not perform such a comparison, as it goes beyond the scope of this work. The dataset obtained in our previous work (ChemPhysChem 2024, e202400547) is available for everybody and the number of molecules there is much larger, than used here (which was limited by the asymmetric tops and availability of experimental rotational constants). And a thorough comparison of consistency between different quantum-chemical approximations would certainly make enough data for a self-sufficient scientific publication. 

Do the authors recommend always using these same tabulated values for a functional/basis-set combination, or would one simply scale as necessary for comparing to a given experiment?

Yes, as in the case of IR spectra, we recommend to use the tabulated values we provide here. See the changes in the text made for issue #1. 

How predictive and generalizable is this across other basis sets and functionals? Are there any reasons it might not work for some functionals (such as the B3LYP data) or even make it worse?

According to illustration cases (Section 4 of the manuscript), this performance is consistent. 

Reviewer 2 Report

Comments and Suggestions for Authors

Report on the manuscript titled "Scaling of rotational constants" by
Denis S. Tikhonov and colleagues.

The authors present a scaling method for the computed rotational constants
at equilibrium geometry, Ae, Be, Ce,  so that they represent as good as possible
the experimental rotational constants A0, B0, C0. Vibrational averaging
to achieve this without scaling seems possible only for smaller molecules.

The scaling procedure, described in section 2, was applied to 117 non-linear
test molecules treated by DFT methods. The results presented in Tables 1 and
2 are somewhat surprising. The message to me seems to be the following:
Just use B3LYP with the def2-TZVP basis set and you are fine. No scaling is
required. Can the authors please comment on this? Did the authors consider,
or plan to consider, ab initio methods such as MP2?

I recommend publication of this manuscript after revision, as it will make an
interesting contribution to the special issue dedicated to Prof. Laane.
Please revise the manuscript taking into account the above remarks.

Author Response

Thank you for the positive evaluation of our work.

The results presented in Tables 1 and 2 are somewhat surprising. The message to me seems to be the following: Just use B3LYP with the def2-TZVP basis set and you are fine. No scaling is required. Can the authors please comment on this?

Yes, it is indeed like that. With B3LYP-D3(BJ)/def2-TZVP and B3LYP-D4/def2-TZVP, no scaling is required. We also found this surprising but obvious in hindsight, as B3LYP with triple-zeta basis sets were always considered to be good for, e.g., assigning the conformers in MW spectra. However, in some sense, this result 1) is the justification of the existing practice amongst the MW spectroscopy community; 2) shows that the B3LYP/def2-TZVP geometries from the experimental point are not exactly the real equilibrium structures, but rather effective r0-structures, which is important to keep in mind upon comparison with other calculations and experimental results. 

Did the authors consider, or plan to consider, ab initio methods such as MP2?

No, we did not consider making MP2-based scaling factors, as these calculations are generally more expensive, and in this work, we were reusing the existing dataset. However, using our procedure and the datasets and procedures from this manuscript and the previous manuscript (ChemPhysChem 2024, e202400547), researchers can obtain scaling factors for any quantum-chemical method of choice. 

Reviewer 3 Report

Comments and Suggestions for Authors

  The authors report on the determination of scaling factors between calculated equilibrium rotational constants and experimental (vibrationally averaged) rotational constants by varying the levels of calculations and basis-sets.  The results are convincing and well suited for the journal Molecules.  Before going directly into the acceptance, the reviewer would like to ask the authors to respond to the following comments and questions.

(1) The difference of the equilibrium rotational constant Be's and the vibrationally averaged one B0's consists of three factors, i. e., harmonic vibration, vibrational anharmonicity, and Coriolis interaction.  It is rather unexpected for the reviewer that one scaling factor can adjust the calculated Be's to B0's fairly well.  Does it mean that the contribution of the harmonic vibrational is dominant and that the other two are marginal?  The authors can quantify these three contributions by using vibrational second order perturbation treatment, as in Gaussian.  Is it possible to carry out the similar calculations using ORCA?

(2) For planer molecules, equilibrium rotational constants must satisfy the relation 1/Ae+1/Be=1/Ce.  For these species, therefore, only two equilibrium constants are independent and can be treated as independent parameters in the fitting.  The reviewer would like to suggest the authors to describe this point explicitly.  

(3) In Table 3, relative mean square deviation of the fitting is significantly worse for HCl(H2O)n species than others.  It may be due to the large anharmonicity of intermolecular vibrations/ internal rotations in such a weakly-bonded species.  That should be addressed in the main text, from the viewpoint of the reviewer.

Author Response

Thank you for the positive evaluation of our work.

(1) The difference of the equilibrium rotational constant Be's and the vibrationally averaged one B0's consists of three factors, i. e., harmonic vibration, vibrational anharmonicity, and Coriolis interaction.  It is rather unexpected for the reviewer that one scaling factor can adjust the calculated Be's to B0's fairly well.  Does it mean that the contribution of the harmonic vibrational is dominant and that the other two are marginal?  The authors can quantify these three contributions by using vibrational second order perturbation treatment, as in Gaussian.  Is it possible to carry out the similar calculations using ORCA?

The contribution of the harmonic vibrations, as it is the symmetric motion, appears only in the J=0 Coriolis contribution, which is negligible and is different from the vibrational bands' position. The main factors controlling the difference of the theoretical rotational constants from the experimental ones, thus, are the anharmonic shift and the systematic error in equilibrium geometry. For our purpose, we do not disentangle these contributions, as we apply only a single scale factor. Quantifying these contributions is a much more computationally challenging problem, as one would require experimental and/or high-quality computed equilibrium structures and the anharmonic shifts for comparison. Therefore, this work is out of the scope of this manuscript. In any case, we would not be able to do such work, as ORCA does not provide the anharmonic shifts for rotational constants, and we do not have a parallelized version of the Gaussian on our computational cluster to perform calculations for large datasets. 

(2) For planer molecules, equilibrium rotational constants must satisfy the relation 1/Ae+1/Be=1/Ce.  For these species, therefore, only two equilibrium constants are independent and can be treated as independent parameters in the fitting.  The reviewer would like to suggest the authors to describe this point explicitly.  

The single scale factor s will not disturb this relation, as (1/Ae+1/Be)/s=1/(sCe).

(3) In Table 3, relative mean square deviation of the fitting is significantly worse for HCl(H2O)n species than others.  It may be due to the large anharmonicity of intermolecular vibrations/ internal rotations in such a weakly-bonded species.  That should be addressed in the main text, from the viewpoint of the reviewer.

To address this point, we have added the following text in the end of section 4:

Note that the rRMSD and MAD deviations for the non-covalently bound cluster are larger than for the covalently bound linear molecules and imidazole isotopologues. This might be either due to the less stiff nature of the intermolecular bonds, which allow for larger systematic deviations of the obtained numerical structures from the actual potential energy surface minimum, or due to a less accurate description of the intermolecular interactions in general, which leads to larger systematic errors, or both.

Reviewer 4 Report

Comments and Suggestions for Authors

The manuscript “Scaling of rotational constants” by Tikhonov et al. translates the established concept of scale factors in computational vibrational spectroscopy to rotational constants. The authors work in the framework of density functional theory and provide numerical data for two popular hybrid exchange-correlation functionals, namely B3LYP and PBE0, as well as two composite methods, PBEh-3c and r2SCAN-3c. Dispersion interactions are consistently considered using D3(BJ) or D4 correction schemes by Grimme et al. The work, although preliminary in nature, can have broad implications for the entire microwave spectroscopy community, especially since the computationally cheap composite methods would allow for efficient prediction of rotational constants.

Overall, I think that this manuscript is certainly publishable in Molecules, provided that the authors respond to the minor questions below:

    1. The quantum chemical software used for the training set calculations is not specified in section 2. I suppose it was Orca, like for the test cases discussed later, but it should be mentioned here as well.

    2. The authors mention “high-quality QC approximations” (line 71), but proceed to present the results using the def2-SVP basis set, which seems deficient for the purpose. As they note later on, the quality of results with a triple-dzeta basis set is higher. The reason to include the fairly minimal basis set should be given.

    3. There are some spurious symbols in line 79 in the text.

    4. In some places in the supplementary spreadsheet, the basis set is specified as “def2-TZVPP”.

    5. The authors’ results for HCl-water clusters indicate that we are still far from very accurate prediction of rotational constants for molecular aggregates with non-covalent interactions. Authors may provide some perspective on that.

Author Response

Thank you for your positive evaluation of our work.

The work, although preliminary in nature, can have broad implications for the entire microwave spectroscopy community, especially since the computationally cheap composite methods would allow for efficient prediction of rotational constants.

The obtained scale factors are sufficiently accurate for the given levels of theory, as characterized by their standard deviations. However, we agree that this work can be expanded to include more quantum-chemical approximations.

  1. The quantum chemical software used for the training set calculations is not specified in section 2. I suppose it was Orca, like for the test cases discussed later, but it should be mentioned here as well.

Yes, it was ORCA, but as calculations were done in previous work (ChemPhysChem 2024, e202400547), we omitted that here. To address this issue, the corresponding description of the dataset in Section 2:

In this case, we chose the set of molecules used for obtaining the scaling factors for harmonic frequencies from Ref. [33], which were computed using ORCA 5 software.[34,35]

  1. The authors mention “high-quality QC approximations” (line 71), but proceed to present the results using the def2-SVP basis set, which seems deficient for the purpose. As they note later on, the quality of results with a triple-dzeta basis set is higher. The reason to include the fairly minimal basis set should be given.

In that particular fragment, we refer to it not as “high-quality” but as “relatively high-quality.” To resolve this misunderstanding, we have modified that sentence to:

Only  the relatively high-quality QC approximations were considered here, namely DF-Dn/def2-mVP (DF = B3LYP, PBE0, n = 3(BJ), 4, m = S, TZ),[36–42] PBEh-3c,[43] and r2SCAN-3c[44] levels of theory, while the calculations with 6-31G basis set were left out from the discussion here.

  1. There are some spurious symbols in line 79 in the text.
  2. In some places in the supplementary spreadsheet, the basis set is specified as “def2-TZVPP”.

Thank you for noticing. It was supposed to be ∆ν/ν and def2-TZVP, respectively. It was corrected now. 

  1. The authors’ results for HCl-water clusters indicate that we are still far from very accurate prediction of rotational constants for molecular aggregates with non-covalent interactions. Authors may provide some perspective on that.

To address this point, we have added the following text at the end of section 4:

Note that the rRMSD and MAD deviations for the non-covalently bound cluster are larger than for the covalently bound linear molecules and imidazole isotopologues. This might be either due to the less stiff nature of the intermolecular bonds, which allow for larger systematic deviations of the obtained numerical structures from the actual potential energy surface minimum, or due to a less accurate description of the intermolecular interactions in general, which leads to larger systematic errors, or both.